# Influence of ZrB_2_ on Microstructure and Properties of Steel Matrix Composites Prepared by Spark Plasma Sintering

**DOI:** 10.3390/ma13112459

**Published:** 2020-05-28

**Authors:** Iwona Sulima, Pawel Hyjek, Lucyna Jaworska, Malgorzata Perek-Nowak

**Affiliations:** 1Institute of Technology, Pedagogical University of Krakow, Podchorazych 2 St., 30-084 Krakow, Poland; pawel.hyjek@up.krakow.pl; 2Faculty of Non-Ferrous Metals, AGH University of Science and Technology, 30-059 Krakow, Poland; ljaw@agh.edu.pl (L.J.); mperek@agh.edu.pl (M.P.-N.)

**Keywords:** steel matrix composite, zirconium diboride (ZrB_2_), spark plasma sintering (SPS/FAST), properties

## Abstract

In this study, four composites with different ZrB_2_ content were made by the Spark Plasma Sintering (SPS/FAST) technique. The sintering process was carried out at 1373 K for 5 min under an argon atmosphere. The effect of ZrB_2_ reinforcing phase content on the density, microstructure, and mechanical and tribological properties of composites was investigated. The results were compared with experimental data obtained for 316L austenitic stainless steel without the reinforcing phase. The results showed that the ZrB_2_ content significantly affected the tested properties. With the increasing content of the ZrB_2_ reinforcing phase, there was an increase in the Young’s modulus and hardness and an improvement in the abrasive wear resistance of sintered composites. In all composites, new fine precipitates were formed and distributed in the steel matrix and along the grain boundaries. Microstructural analysis (Scanning Electron Microscopy (SEM), Wavelength Dispersive Spectroscopy (WDS)) has revealed that the fine precipitates chromium contained chromium as well as boron.

## 1. Introduction

Powder metallurgy is the most attractive producing route for these types of particulate-reinforced metal-matrix composites. The use of powder metallurgy for the manufacture of metal matrix composites allows obtaining a diffusive bond between the matrix and the reinforcing phase. It also creates a vast range of possibilities as regards the choice of the type, form and size of the reinforcing phase and enables making products which, while having the same chemical composition, are characterized by different densities [1,2,3]. Many technologies including free sintering [4,5], self-propagating high-temperature synthesis (SHS) [6,7], Hot Isostatic Pressing (HIP) [8,9] and Spark Plasma Sintering (SPS/FAST) [10] have been used to make steel matrix composites. The SPS/FAST method is based on the simultaneous application of pressure and pulsed current [11,12]. Compared to conventional sintering methods, the use of pulsed current increases the sintering kinetics and allows sintering in a lower range of temperature and pressure values. Other advantages of the SPS/FAST method include the elimination of sintering activators and material consolidation without the need for pre-pressing, isostatic compaction and drying [13,14,15].

Iron-based alloys are very often used as a matrix of metal-based composite materials, this is mainly due to the low manufacturing cost, high mechanical properties and satisfactory wear resistance [16,17,18]. The 316L austenitic steel is characterized by very high mechanical properties and equally high resistance to the effect of aggressive environments [16]. The introduction of hard ceramic particles into the steel matrix improves the mechanical and tribological properties of such composites [19,20]. The ceramic particles used most commonly as a reinforcement of the steel matrix are oxides [21,22,23], carbides [24,25], nitrides [26] and borides [10,27,28,29]. Most research concerns optimization of the sintering process, determination of the impact of different sintering conditions and reinforcing phase content on the physical (density, porosity), mechanical and microstructural properties of composites. The corrosion behavior and tribological properties of steel-based composites reinforced with ceramic particles are also investigated [30,31,32].

Among various ceramic particulates, zirconium diboride is considered one of the best reinforcements of the steel matrix mainly due to its high melting temperature (3246 °C), low density (6.085 g/cm^3^) and high tribological properties. ZrB_2_ has good mechanical properties like hardness (>2200 HV), Young’s modulus (450 GPa) and excellent electrical (1 × 10^7^ S/m) and thermal conductivity (60 W/m K) [33,34,35]. Zirconium diboride ceramics have excellent corrosion resistance against molten iron and its alloys and high thermal shock resistance [36].

Research has been conducted on various composite systems such as Al/ZrB_2_ [37,38], AA7075/ZrB_2_ [39], AA6061/ZrB_2_ [40], Cu/ZrB_2_ [41], Al-Cu/ZrB_2_ [42], and reported the conclusions regarding physical, mechanical and tribological properties. However, the literature contains only scarce information on studies concerning the manufacture of ZrB_2_-reinforced, iron alloy-based composites by modern technologies using pulsed current. A review of the literature has indicated that issues related to the sintering process of composites based on iron alloys reinforced with another boride, i.e., TiB_2_, were the subject of several research works [8,10,43,44,45,46]. Sulima et al. studied the effect of varying amounts of TiB_2_ (10 vol.% and 20 vol.%) on the mechanical and tribological properties of composites based on 316L austenitic steel. It was shown that composites reinforced with titanium diboride are characterized by high Young’s modulus, hardness and compressive strength. The friction coefficient of composites is decreasing with the increasing content of TiB_2_ in the composite matrix. The best tribological properties were obtained for the composite with 20 vol.% TiB_2_. In another research work [8], microstructure and mechanical properties were tested, including high temperature testing of 316L steel matrix composites reinforced with 15 vol.% TiB_2_. The test materials were manufactured by the HIP sintering process. Examinations of the composite microstructure showed a homogeneous distribution of TiB_2_ in the steel matrix. The authors demonstrated that the introduction of fine TiB_2_ ceramic particles into the steel matrix is a good way to improve the mechanical properties of composite materials. Tjong and Lau [47] attempted to produce by HIP a composite based on 304 austenitic steel containing 20 vol.% TiB_2_. The results showed that the addition of 20 vol.% TiB_2_ to 304 steel significantly improves the wear resistance of composites.

The main objective of the present study is to determine the effect of content of the reinforcing ZrB_2_ phase on density, microstructure and properties of composites based on 316L austenitic stainless steel.

## 2. Materials and Experimental Methods

The materials used in this study were the 316L austenitic stainless steel powder (Hoganas, about 25 μm average grain size) and ZrB_2_ powder (H.C. Starck, 2.5–3.5 μm average grain size, purity 99.9%). The chemical composition of the austenitic stainless steel powder was 17.20 wt.% Cr, 12.32 wt.% Ni, 2.02 wt.% Mo, 0.43 wt.% Mn, 0.89 wt.% Si, 0.03 wt.% S, 0.028 wt.% P, 0.03 wt.% C and balance of Fe. Figure 1 presents morphology of 316L steel and ZrB_2_ powders. The powders’ mixtures (Figure 2) were prepared by mixing in a turbula mixer for 8 h. The following compositions were prepared:316L steel + 5 wt.% ZrB_2_316L steel + 10 wt.% ZrB_2_316L steel + 15 wt.% ZrB_2_316L steel + 20 wt.% ZrB_2._

For comparison, powder of commercial 316L austenitic stainless steel was sintered.

The materials were sintered using Spark Plasma Sintering (SPS/FAST). The SPS machine type HPD 5 (FCT System, Frankenblick, Germany) was used. Powders were poured into a graphite die of 20 mm inner diameter, and the die was placed between the graphite electrodes in an SPS chamber. The process of compaction was conducted in vacuum under a maximum pressure of 35 MPa. The maximum pressure was obtained after 10 min of the test duration. Both the vacuum and the pressing time of 10 min were applied in order to “vent” the mixture. After this step, argon acting as a protective gas was introduced into the SPS furnace chamber, and the sintering process continued. The powders were sintered at 1373 K for 5 min. The heating and cooling rate of the furnace were kept 473 K/min and 373 K/min, respectively. During the SPS/FAST process, the temperature was monitored by a pyrometer. Cylindrical sintered samples with dimensions of 20 mm in diameter and 8 mm thickness were produced. Before examination, the surfaces of the sintered composites were machined to remove the layer contaminated by the carbon sheet. The samples were disc-shaped. The measurements were realized with specimen of 20 mm in diameter and 8 mm in high. Ion etchings of the samples were carried out using the apparatus of PECS manual Gatan (Gatan, Pleasanton, CA, USA).

Samples for microstructural analysis were prepared by standard methods of grinding using SiC foil and of polishing up to 1 μm using diamond suspension and MD-Dac discs. Ion etchings of the samples were carried out using the apparatus of PECS manual Gatan. The microstructure of the sintered materials was evaluated by Scanning Electron Microscopy (SEM) JEOL JSM 6610LV (Tokyo, Japan) with Energy Dispersive Spectroscopy EDS (AZtec) and Hitachi (Tokyo, Japan) SU-70 with Wavelength Dispersive Spectroscopy (WDS). The phase analysis was carried out by X-ray diffraction technique (Empyrean; PANalytical, Almelo, The Netherlands) using CuKα radiation.

The density of the sintered composites was determined by the Archimedes’ immersion method in water [48]. A systematic error of the density measurements in samples weighing more than 1 g was below 0.1%. Young’s modulus of the composites was measured based on the velocity of the ultrasonic wave transition through the sample using ultrasonic flaw detector Panametrics Epoch III (Billerica, MA, USA). The accuracy of calculated Young’s modulus was estimated at 2%. For each sample, five measurements of Young’s modulus were carried out. Microhardness of the sintered samples were carried out using Vickers hardness tester (NEXUS 400, Falls Church, VA, USA) with indentation loads of 2.94 N. Eight indentations were realized for each sample.

Tribological tests were performed using a ball-on-disc wear testing machine under a load of 5 N, sliding speed of 0.1 m/s and at a distance of 1000 m. The test duration and friction track diameter were 10,000 s and 5 mm, respectively. The tribological tests were conducted using two types of counterface balls (steel AISI52100 and Al_2_O_3_). The ball had a diameter of 3.175 mm. For each test a new ball was used. Specimens were washed in high purity acetone and dried. After the ball and sample were mounted, materials were washed in ethyl alcohol and then dried. The surface roughness was below 0.2 μm (R_a_). The experimental procedure followed the ISO 20808:2004(E) [49]. The values of friction coefficient were calculated from the following equation:(1)$$µ=\frac{F_{f}}{F_{n}L}$$
where:*F_f_* is the measured friction force,*F_n_* is the applied normal force,*L*—sliding distance [m].

Specific wear rate was calculated by means of equation: (2)$$W_{V\left(disc\right)}=\frac{V_{disc}}{F_{n}\cdot L}$$
where:*W_V_*_(*disc*)_**—specific wear rate of disc [mm^3^/Nm]; *V_disc_*—wear volume of disc specimen [mm^3^]; *F_n_*—applied load [N]; *L*—sliding distance [m].

The wear volume of disc specimen was calculated from following equation:(3)$$V_{disc}=\frac{Ⅱ}{2}\cdot R\cdot \left(S_{1}+S_{2}+S_{3}+S_{4}\right)$$
where:*R*—radius of wear track [mm], *S*_1_ to *S*_4_**—cross-sectional areas at four places on the wear track circle [mm^2^].

## 3. Results

Figure 3 shows the X-ray diffraction patterns of 316L steel and composites containing 5–20% of the reinforcement. 

Analysis of the phase composition showed the presence of only the ZrB_2_ phase and austenitic steel. It can be easily noticed that the intensity of ZrB_2_ peaks increases with the increasing amount of ZrB_2_ particles in the composite. The SEM micrographs of the 316L steel and composites containing 5–20% ZrB_2_ are presented in Figure 4a–e, respectively. 

The micrographs show the morphology and distribution of ZrB_2_ particles in a stainless steel matrix. The size of particles in this reinforcement ranges from 1 to 6 μm. Zirconium diboride tends to be located along the boundaries of the steel matrix grains. The ZrB_2_ phase was found to be uniformly distributed in the sintered composites. With an increase in the weight fraction, the reinforcing phase formed a continuous layer along the matrix grain boundaries. Locally, agglomeration of the ZrB_2_ reinforcing phase was observed (e.g., in Figure 4c). 

Microstructural examinations revealed several characteristic features of the microstructure of sintered composites containing 5–20% ZrB_2_. The formation of a large number of small precipitates distributed in a 316L steel matrix was observed (Figure 5, Figure 6, Figure 7 and Figure 8). Locally, the cracks on the surface of the reinforcing phase were observed, which might be due to the effect of pressure applied in the sintering process (Figure 8). Figure 5 shows an example of the microstructure of composites reinforced with zirconium diboride. In the steel matrix grains and along the grain boundaries, the scattered dispersion precipitates of a chromium-containing phase are visible (analysis of points No. 3, Figure 5). 

Such precipitates were not observed in the sintered 316L steel. As revealed by the WDS analysis of chemical composition (Figure 6, Figure 7 and Figure 8), the precipitates contained chromium as well as boron. Therefore, it seems probable that during the sintering process, new borides might form in the microstructure of the steel–ZrB_2_ composite. Various mechanisms accompanying the SPS/FAST process, like surface activation, diffusion, partial melting, neck formation between sintered powder particles and plastic flow, could influence this phenomenon [11,15]. Additionally, in the microstructure of all steel–ZrB_2_ composites, local formation of nickel-rich precipitates, which tended to be located near the ZrB_2_ reinforcing phase (Figure 8), was observed. The authors of Ref. [50] demonstrated that the microstructure of a sintered steel matrix composite depends on the sintering method used. Consequently, applying two sintering techniques, i.e., HP-HT and SPS/FAST, a steel matrix composite containing 8 vol.% TiB_2_ was produced. Studies of the microstructure revealed some differences in the phase composition of sintered steel–TiB_2_ composites. The presence of chromium–iron–nickel phase (Cr_0.18_Fe_0.09_Ni_0.73_) and locally occurring nickel-containing precipitates were detected in the composites sintered by HP-HT. Composites made by the SPS/FAST method contained the Cr_0.18_Fe_0.09_Ni_0.73_ phase and two complex borides. In turn, Molinari and al. [51,52] described the formation of complex borides (Cr, Mo, Fe)_2_B in the free sintering process of austenitic steel with the use of boron as a sintering activator. In this case, the sintering process took place in the liquid phase, which was formed as a result of eutectic reaction between the steel matrix and the boride phase ((Fe, Cr, Mo)_2_B).

Table 1 presents the results of studies of the physical properties obtained in sintered 316L steel and composites with different ZrB_2_ content. 

All the materials sintered by the SPS/FAST method were characterized by a high degree of consolidation. For these materials, a high density reaching 93–98% of the theoretical density was obtained. The results show that the apparent density of the composite with 20% ZrB_2_ is the lowest and amounts to 7.07 g/cm^3^. This indicates a decrease in density with the increasing content of ZrB_2_ in the steel matrix (rys.9) and is the phenomenon which might be easily expected, since the density of zirconium diboride (6.085 g/cm^3^) is lower than the density of steel (8.0 g/cm^3^). All composites have a low open porosity of 0.17%, 0.85%, 1.24% and 3.12% for the ZrB_2_ content of 5%, 10%, 15% and 20%, respectively.

As follows from the analysis of test results, the Young’s modulus increases with an increase in the reinforcing phase content (Table 1, Figure 9). The highest value of Young’s modulus (236 GPa) was obtained for the composites containing 20% ZrB_2_. For comparison, for the sintered 316L steel without reinforcement, the obtained value of Young’s modulus was 204 GPa. Figure 10 shows the results of microhardness tests as a function of changes in the ZrB_2_ reinforcing phase content in the matrix. With the increasing content of ZrB_2_, the microhardness also increases, and from the value of 183 HV0.3 obtained for 316L steel without reinforcement, it increases more than twice (up to 440 HV0.3) for composites containing 20% ZrB_2_.

Abrasion resistance tests were carried out at room temperature in a ball-on-disc system. Two balls made from the AISI52100 steel and Al_2_O_3_ were used as counter-samples. Changes in the friction coefficient as a function of the ZrB_2_ content are shown in Figure 11. All curves change in a similar way. The curves of the composites with a higher content of ZrB_2_ (over 15%) are more stable. All curves show the break-in period at the beginning of the abrasion test. The results of abrasive wear resistance tests demonstrate that the coefficient of friction depends on the ZrB_2_ reinforcing phase content in a steel matrix. The coefficient of friction decreases with the increasing ZrB_2_ content. The lowest coefficient of friction (0.4) was recorded for steel–20% ZrB_2_ composites.

The next step was to find out what effect the type of counter-sample had on the friction coefficient (Figure 11a or Figure 12a). Lower friction coefficients were obtained in sintered materials tested with the use of the Al_2_O_3_ counter-samples. For these test conditions, the coefficient of friction ranged from 0.4 to 0.59 and depended on the amount of the ZrB_2_ reinforcing phase. For comparison, the use of a steel counter-sample in abrasion tests gave higher values of the friction coefficient, i.e., in the range of 0.56–0.64. These results are consistent with the results obtained by Wang et al. [53], who studied the tribological properties of mesocarbon microbead–silicon carbide (MCMBs–SiC) composites. Tribological tests were conducted on a standard block-on-ring friction testing machine. Three kinds of counterparts were involved in the friction tests: MCMBs–SiC composites which were made through the same method with the blocks, sintered SiC and WC. The results showed that the coefficient of friction and wear rate of MCMBs–SiC composites was related to the coefficient of friction and mostly depended on the counterparts. Kovalčíková et al. [54] investigated the tribological behavior of SiC materials in contact with Si_3_N_4_, Al_2_O_3_, and ZrO_2_ ceramic balls and a WC–Co ball. Wear testing was carried out using the ball-on-disc technique. The results were concluded that the coefficient of friction is strongly dependent on testing conditions and further by the counterpart material.

Compared to steel without reinforcement, the introduction of ZrB_2_ into a steel matrix reduces the wear rate of composites (Figure 12b,c). Both weight loss and specific wear rate strongly depend on the reinforcing phase content in the composite, and this is mainly due to the presence of hard ceramic particles in the steel matrix. According to the literature, the hardness of zirconium diboride is about 2200 HV [33,34,35]. 

Hence, the hardness of composites tends to increase as the ZrB_2_ content increases, and the wear resistance is improved. For sintered 316L steel tested with an Al_2_O_3_ counter-sample, the weight loss and specific wear rate were 2.3 g and 8030 × 10^−6^ mm^3^/N m, respectively. For comparison, the 20% ZrB_2_ composite materials tested with an Al_2_O_3_ counter-sample showed the weight loss and specific wear rate of 0.44 g and 1892 × 10^−6^ mm^3^/N m, respectively. The same trend was observed in abrasion tests using a steel counter-sample. Hence, the conclusion follows that zirconium diboride particles protect the steel matrix during friction and reduce its wear. With the increasing content of ZrB_2_, the loss of composite material is less severe. Examining carefully the effect of the type of counter-sample on the wear rate of the tested materials, it can be noticed that higher values of the wear index were obtained for both 316L steel and composite materials during abrasion tests using Al_2_O_3_ balls (Figure 12c). For the sintered steel without reinforcement and composites with 5–20% ZrB_2_, the values of the specific wear rate were 40% higher than the results of the tests carried out with steel balls. Changes in the weight loss, studied as a function of the type of counter-sample used, followed the same tendency. The differences in the obtained values of the mass loss and specific wear rate may result from differences in the hardness of materials used for counter-samples. According to the literature data, the hardness of Al_2_O_3_ and AISI52100 steel is 1400–1800 HV and 850 HV [55,56], respectively. The Al_2_O_3_ ball is harder, so it can more easily penetrate and remove material from the tested surface.

After tribological tests, the wear tracks were examined in the sintered materials. Examples of wear tracks obtained with the use of Al_2_O_3_ counter-sample are shown in Figure 13. Comparing the wear tracks, some characteristic features of abrasive and adhesive wear, such as scratches and furrows arranged in the direction of ball movement (the bright arrow on microstructures) and delaminations, were observed on the surface. During the tests, permanent deformation of the sintered materials combined with abrasion took place in the wear track. Characteristic uneven seizures and local areas where the tested material was removed from the wear track were observed. Microstructural examinations showed that for the tested materials, the width of the track was decreasing with the ZrB_2_ content increasing in the steel matrix. The largest width of the wear track was found in 316L steel without reinforcement (Figure 13a) and it was gradually decreasing compared to the width of the wear track observed successively in composites with 5–20% ZrB_2_, (Figure 13b–e). These results confirm that the Al_2_O_3_ ball used as a counter-sample abrades more easily the surface of the steel sample without reinforcement, removing a larger volume of material. Hence it follows that the ZrB_2_ reinforcing phase protects the austenitic steel matrix during the friction process and reduces its wear. 

## 4. Conclusions

Austenitic stainless steel-based composites with different ZrB_2_ content can be successfully manufactured by the SPS/FAST method. The apparent density of sintered composites was above 93% of theoretical density. Examinations of the composite microstructure revealed a homogeneous distribution of the ZrB_2_ reinforcing phase in a steel matrix. In all composites, new fine precipitates were formed and distributed in the steel matrix and along the grain boundaries.

Test results showed a significant effect of the amount of zirconium diboride on the properties of composites. An increase in the weight fraction of ZrB_2_ caused an increase in the Young’s modulus and microhardness. For composites containing 20% ZrB_2_, the microhardness was more than twice as high as for 316L steel without reinforcement. The introduction of the ZrB_2_ into the steel matrix had a beneficial effect on the wear resistance of composites. With the increasing content of ZrB_2_, the friction coefficient, weight loss and specific wear rate were decreasing. The best combination of mechanical and tribological properties was obtained for the steel matrix composites reinforced with 20% ZrB_2_.

## Figures and Tables

**Figure 1 materials-13-02459-f001:**
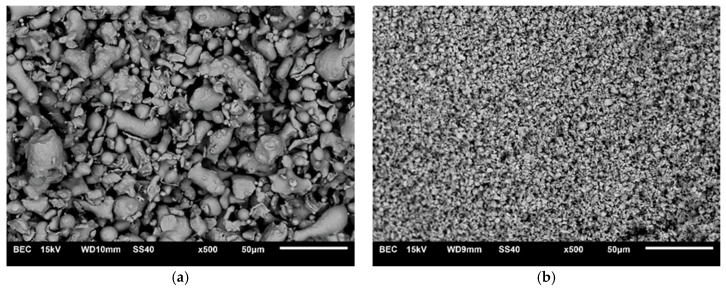
Scanning Electron Microscopy (SEM) images of (**a**) 316L steel and (**b**) zirconium boride (ZrB_2_) powders.

**Figure 2 materials-13-02459-f002:**
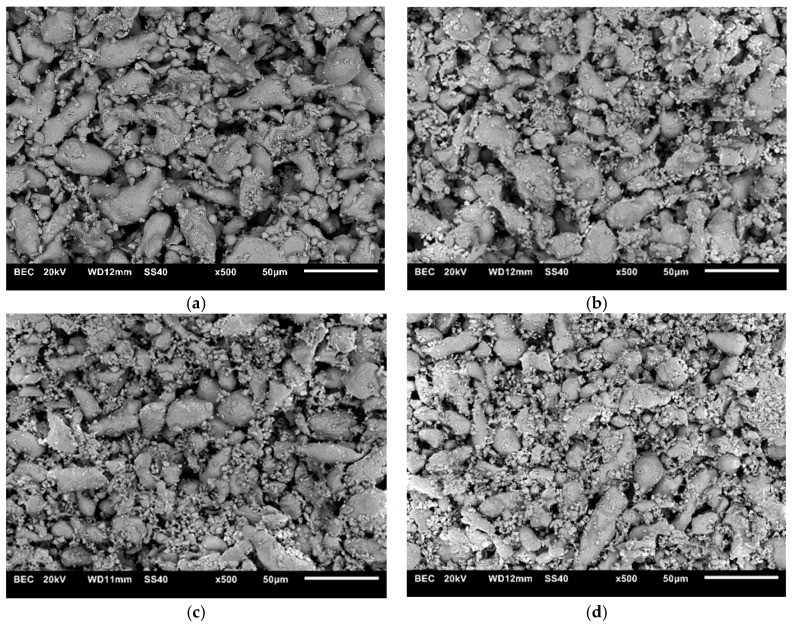
SEM images of powders: (**a**) steel + 5% ZrB_2_, (**b**) steel + 10% ZrB_2_, (**c**) steel + 15% ZrB_2_ and (**d**) steel + 20% ZrB_2_.

**Figure 3 materials-13-02459-f003:**
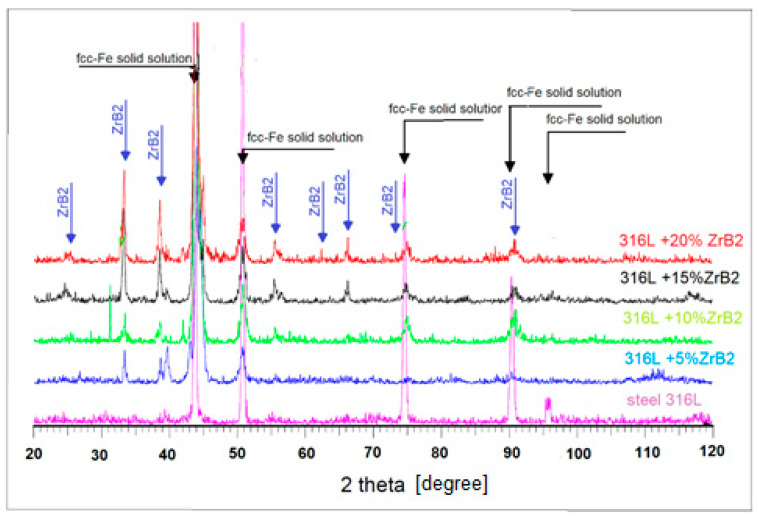
Diffraction pattern of sintered materials obtained by SPS at 1373 K under 35 MPa and during 5 min.

**Figure 4 materials-13-02459-f004:**
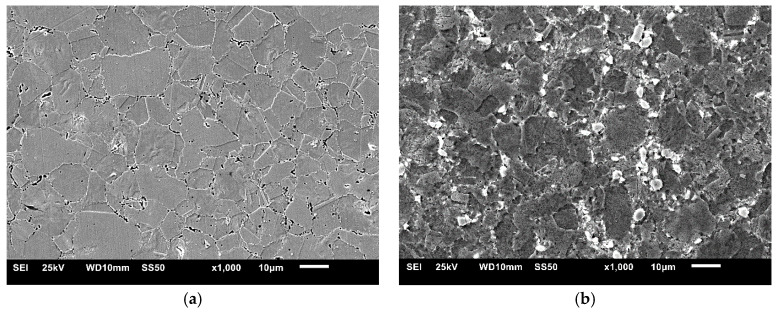
The microstructure (SEM) of (**a**) 316L steel and composites containing (**b**) 5% ZrB_2_, (**c**) 10% ZrB_2_, (**d**) 15% ZrB_2_ and (**e**) 20% ZrB_2_.

**Figure 5 materials-13-02459-f005:**
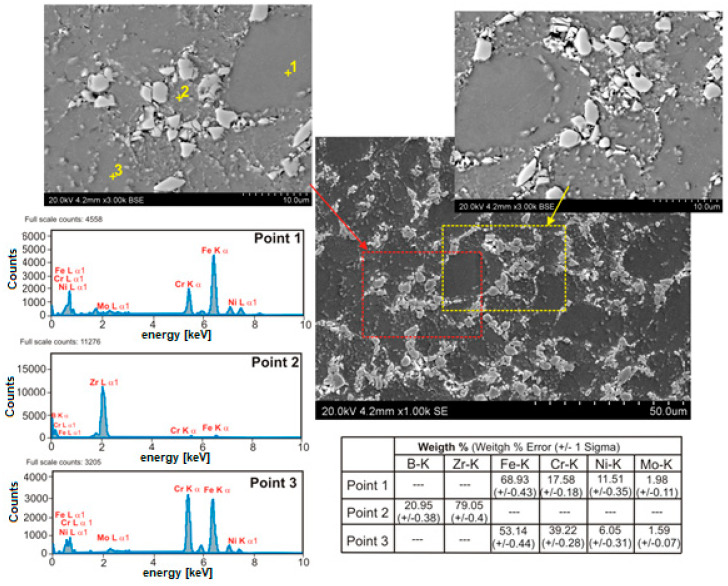
The microstructure (SEM) of steel +20% ZrB2 with corresponding area analysis (EDS).

**Figure 6 materials-13-02459-f006:**
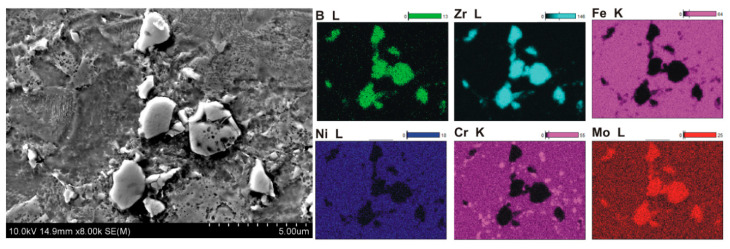
Microstructure and element surface distribution (Wavelength Dispersive Spectroscopy WDS) of steel +5%ZrB_2_ composites.

**Figure 7 materials-13-02459-f007:**
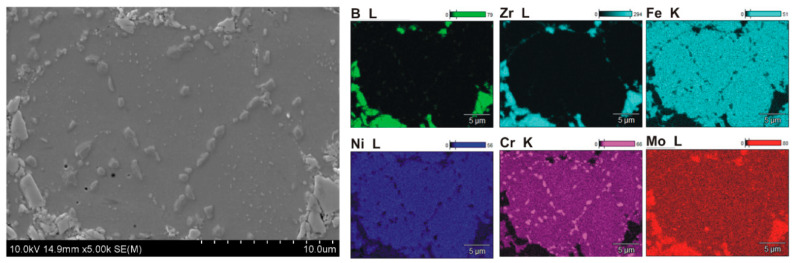
Microstructure and element surface distribution (WDS) of steel +20% ZrB_2_ composites.

**Figure 8 materials-13-02459-f008:**
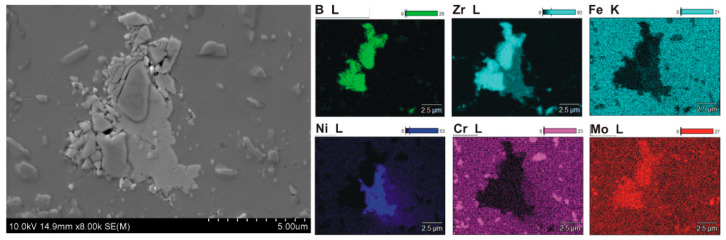
SEM image of steel +20% ZrB_2_ composites showing nickel-rich precipitate with element surface distribution (WDS).

**Figure 9 materials-13-02459-f009:**
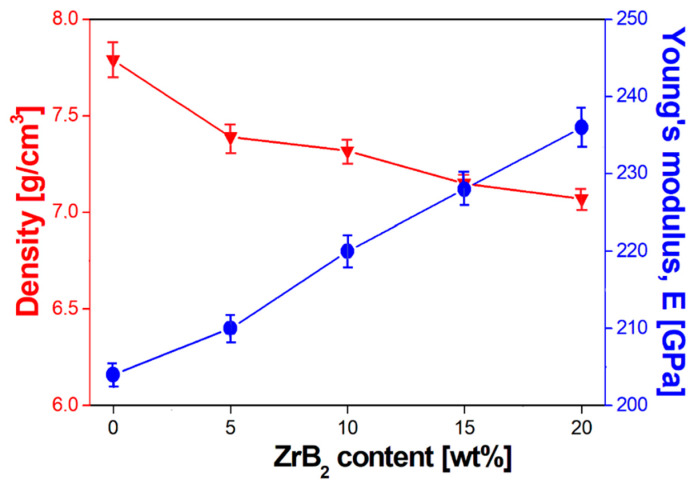
Effect of ZrB_2_ content on density and Young’s modulus.

**Figure 10 materials-13-02459-f010:**
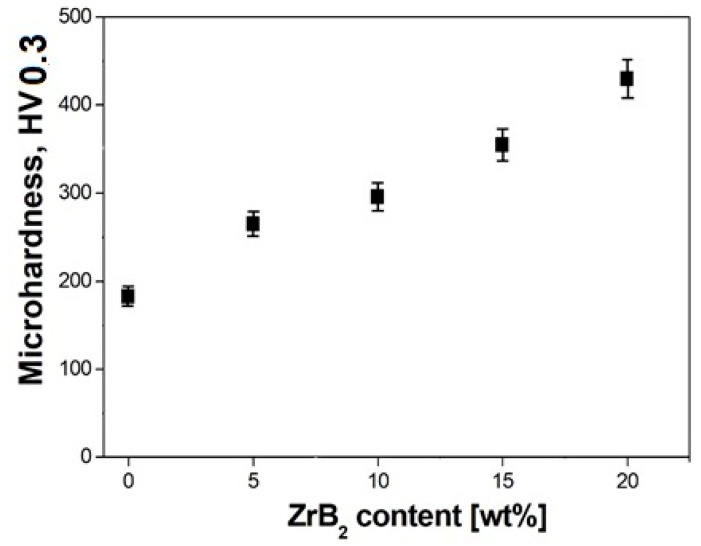
Effect of ZrB_2_ content on microhardness.

**Figure 11 materials-13-02459-f011:**
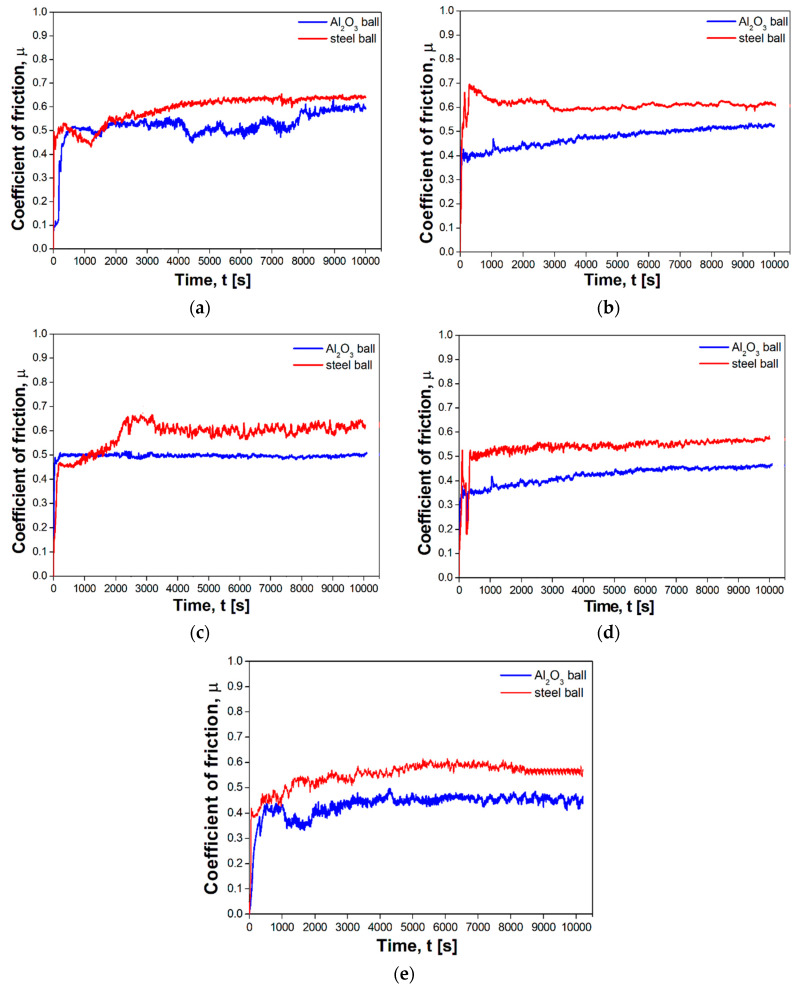
Typical COF curves of (**a**) 316L steel and composites containing: (**b**) 5% ZrB_2_, (**c**) 10% ZrB_2_, (**d**) 15% ZrB_2_ and (**e**) 20% ZrB_2_ as function of testing time, measured using different balls.

**Figure 12 materials-13-02459-f012:**
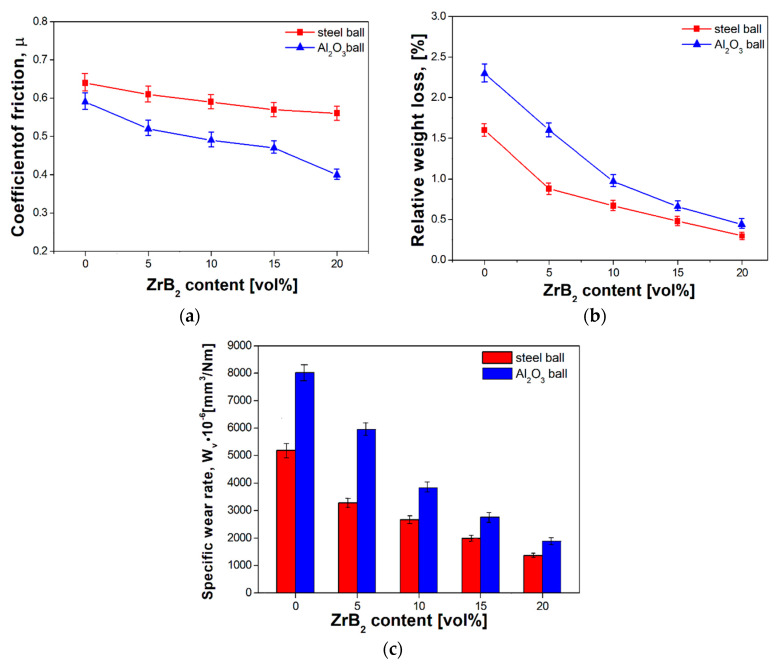
Effect of ZrB_2_ content on (**a**) coefficient of friction, (**b**) relative weight and (**c**) specific wear rate of sintered composites.

**Figure 13 materials-13-02459-f013:**
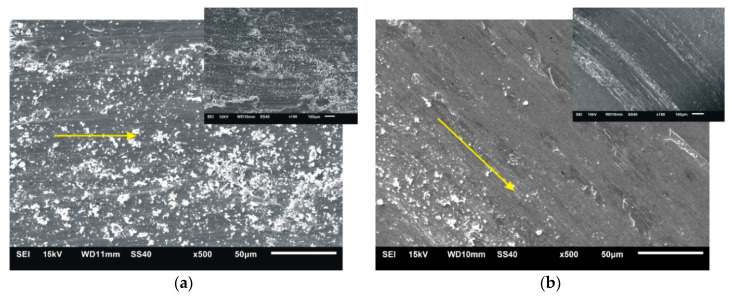
Worn surfaces after the wear test with applied Al_2_O_3_ ball for: (**a**) 316L steel, and (**b**) steel + 5% ZrB_2_, (**c**) steel + 10% ZrB_2_ (**d**) steel + 15% ZrB_2_ and (**e**) steel + 20% ZrB_2_ composites.

**Table 1 materials-13-02459-t001:** The properties of 316L steel and steel–ZrB_2_ composites sintered by SPS/FAST method.

Sintered Materials	Apparent Density *ρ* [g/cm^3^]	$\frac{\rho_{o}}{\rho_{Teo}}$ [%]	Poisson’s Ratioν	Young’s Modulus E [GPa]	$\frac{E}{E_{Teo}}$ [%]
316L steel	7.79 ± 0.02	98	0.28	204 ± 4	91
Steel + 5% ZrB_2_	7.39 ± 0.02	94	0.28	210 ± 4	88
steel + 10% ZrB_2_	7.32 ± 0.02	94	0.27	220 ± 4	87
steel + 15% ZrB_2_	7.15 ± 0.02	93	0.26	228 ± 5	90
steel + 20% ZrB_2_	7.07 ± 0.02	93	0.27	236 ± 5	89

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
