# Peer review of "Influence of ZrB2 on Microstructure and Properties of Steel Matrix Composites Prepared by Spark Plasma Sintering"

_materials, 2020, doi:10.3390/ma13112459_

Round 1

Reviewer 1 Report

This work present the fabrication, structural characterizaion and property evaluation of steel-ZrB2 composites obtained by field-assisted sintering. The work is overall of good quality and detailed enough to be of both scientific and practical interest to the materials science community.

I suggest that some changes be made before publication.

Please clarify the statement in the Abstract "In all composites, new fine precipitates were formed and distributed in the steel matrix and along the grain boundaries" - what are those new precipitates? This is explained in the text but would be good to see this information in the Abstract as well.

Why was SPS conducted in argon and not in vacuum?

Please improve the picture quality of Fig. 3. When you mark the peaks, would it not be better to use "fcc-Fe solid solution" rather than "alloy"? "Alloy" is not always a single phase.

Why was ball milling not considered for making the composite powders? The structure of sintered materials could have been tailored by changing the structure of the initial mixture.

Could you include a SEM image of the sintered steel (without ZrB2) for the sake of comparison of the structure with that of the composites?

Do you attribute the formation of "new" fine precipitates to the specifics of the sintering method used or to the presence of ZrB2? Is it possible to write a chemical reaction (reaction scheme) describing the process of the formation of Ni-rich and Cr-rich precipitates?

Thank you for your interesting work.

Author Response

1. Please clarify the statement in the Abstract "In all composites, new fine precipitates were formed and distributed in the steel matrix and along the grain boundaries" - what are those new precipitates? This is explained in the text but would be good to see this information in the Abstract as well.

It was changed.

2. Why was SPS conducted in argon and not in vacuum?

This text were added to the manuscript.

They were poured into a graphite die of 20mm inner diameter, and the die was placed between the graphite electrodes in a SPS chamber. The process of compaction was conducted in vacuum under a maximum pressure of 35 MPa. The maximum pressure was obtained after 10 minutes of the test duration. Both the vacuum and the pressing time of 10 minutes were applied in order to “vent” the mixture. After this step, argon acting as a protective gas was introduced into the SPS furnace chamber, and the sintering process continued.

3. Please improve the picture quality of Fig. 3. When you mark the peaks, would it not be better to use "fcc-Fe solid solution" rather than "alloy"? "Alloy" is not always a single phase.

The image was change.

4. Why was ball milling not considered for making the composite powders? The structure of sintered materials could have been tailored by changing the structure of the initial mixture.

The authors conduct the investigations on the preparation of steel-ZrB2 mixtures using ball mill. Such mixtures are currently sintering using SPS/FAST. These studies will be the subject of a new article.

5. Could you include a SEM image of the sintered steel (without ZrB2) for the sake of comparison of the structure with that of the composites?

The image was added.

6. Do you attribute the formation of "new" fine precipitates to the specifics of the sintering method used or to the presence of ZrB2? Is it possible to write a chemical reaction (reaction scheme) describing the process of the formation of Ni-rich and Cr-rich precipitates?

The microstructure of a sintered steel matrix composite can depends on the sintering method used. In other studies, the authors was produced a steel matrix composite containing 8 vol% TiB2 applying two sintering techniques, i.e. HP-HT and SPS/FAST. Microscopic examinations have revealed a close dependence of microstructure on the methods and conditions of sintering. This results were published in publication [53] (Sulima I.,  Boczkal S., Jaworska L., SEM and TEM characterization of microstructure of stainless steel composites reinforced with TiB2, Materials Characterization, 118, 2016, 560-569.). In the steel-8TiB2 composites sintered by the method of SPS, the formation of complex borides, and of the FeCr phase and Cr0.18Fe0.09Ni0.73 phase was observed. Different sintering times used in the SPS method mainly affected the number and size of complex borides. Extending the time to 30 minutes resulted in an increase and growth of borides. 

Reviewer 2 Report

The manuscript presented detailed background introduction, experimental and results of steel + ZrB2 alloy systems synthesized by using SPS technique. The results can be useful as references when such alloy systems are needed in an application.

  1. 4, two pictures are labeled as (b).
  2. Both Figures 7 and 8 are for steel + 20% ZrB2. What’s the reason of showing two figures of the same alloy but omitting 10% and 15% ZrB2 alloys?
  3. 11, two pictures are labeled as (b). Caption is not detailed to illustrate which picture is for which alloy.
  4. 12, missing label (c).

Author Response

1)   4, two pictures are labeled as (b).

It  was changed.

2)  Both Figures 7 and 8 are for steel + 20% ZrB2. What’s the reason of showing two figures of the same alloy but omitting 10% and 15% ZrB2 alloys?

For comparison, the article contains only selected microstructures of composites with 5 and 20% ZrB2 (smallest and largest content). For composites with 10% and 15% ZrB2, the morphology and chemical composition were similar. Also, the scattered dispersion precipitates of a chromium-containing phase were visible

3) 11, two pictures are labeled as (b). Caption is not detailed to illustrate which picture is for which alloy.

It was changed.

4) 12, missing label (c).

It was changed.

Reviewer 3 Report

Dear authors,

 I have found Your article well prepared, with scientific soundness and good level of English. I was really surprised, The WoS find just one reference to combination 316 and ZrB2! There was omitted couple of mistakes:

P6, Fig. 5 The quality of figure is low and it is not possible to read inlets properly. Try to increase resolution of inlets from EDS software and whole figure to increase its readability.

 P7, l. 168: I would prefer: The authors of Ref. [53]….

P10, Fig. 11 There is missing the description of particular sub-figures!!! My main complain, add it

P11, l. 250 Do not use asterisk * for multiplication, use character ⋅ (dot in the middle of line). The same P11, l. 252

Good luck!

Author Response

1)  P6, Fig. 5 The quality of figure is low and it is not possible to read inlets properly. Try to increase resolution of inlets from EDS software and whole figure to increase its readability.

The figures were changed.

 2)  P7, l. 168: I would prefer: The authors of Ref. [53]….

It was changed.

3)  P10, Fig. 11 There is missing the description of particular sub-figures!!! My main complain, add it

The figures were changed.

4)  P11, l. 250 Do not use asterisk * for multiplication, use character ⋅ (dot in the middle of line). The same P11, l. 252

It was changed.

Reviewer 4 Report

  • Heating and Cooling rates always in Kelvin
  • different grain sizes factor of 10 will not cause separation during milling?
  • is the surface denser during SPS? Was it removed for the further measurements as density ?
  • If using a carbon crucible for SPS (there are no information on the crucible) there might be a reaction with the steel. Have the authors considert to this?
  • Figures are of bad quality in particular the XRD one.
  • Figure 4 are facture images? it would be better to have here polished cross sections
  • Figure 5 bad quality,what type do we see? Fracture? Surface? Grinded / polished?
  • MIcrostructure of 0% ZrB2 is missing to see the difference with 5% ZrB2
  • Table 1: the % column describe what?
  • Figure 9 : the error bars are missing
  • Figure 11: What is described by a-e) there are no information which composition is tested
  • Figure 12: where do the values come from? There are no error bars. If they were taken from the experiments there should be error bars

The manuscript lacks from a lot of missing details describing the research.

Author Response

1)  Heating and Cooling rates always in Kelvin

It was changed

2) different grain sizes factor of 10 will not cause separation during milling?

The authors conduct the investigations on the preparation of steel-ZrB2 mixtures using ball mill. Such mixtures are currently sintering using SPS/FAST. These studies will be the subject of a new article.

3) is the surface denser during SPS? Was it removed for the further measurements as density If using a carbon crucible for SPS (there are no information on the crucible) there might be a reaction with the steel. Have the authors considert to this?

Before all examination ( tests of density, Young’s modulus etc.), the surfaces of the sintered composites were machined to remove the layer contaminated by the carbon sheet. The samples were disc-shaped. The measurements were realized with specimen of 20 mm in diameter and 8 mm in high.

4) Figures are of bad quality in particular the XRD one.

The image was changed

5)  Figure 4 are facture images? it would be better to have here polished cross sections

Samples for microstructural analysis were prepared by standard methods of grinding using SiC foil and of polishing up to 1 mm using diamond suspension and MD-Dac discs. Ion etching of the samples were carried out using the apparatus of PECS manual Gatan.

6) Figure 5 bad quality,what type do we see? Fracture? Surface? Grinded / polished?

Samples for microstructural analysis were prepared by standard methods of grinding using SiC foil and of polishing up to 1 mm using diamond suspension and MD-Dac discs. Ion etching of the samples were carried out using the apparatus of PECS manual Gatan.

7) MIcrostructure of 0% ZrB2 is missing to see the difference with 5% ZrB2

The image was added.

8) Table 1: the % column describe what?

Information was added.

9) Figure 9 : the error bars are missing

The Figure was changed

10) Figure 11: What is described by a-e) there are no information which composition is tested

It was changed.

11) Figure 12: where do the values come from? There are no error bars. If they were taken from the experiments there should be error bars no

The error bars were added

Round 2

Reviewer 4 Report

Fig. 7 and 8. same description where is the difference?

ZrB2 no subscript of 2 in all figure descriptions also for Al2O3

Fig. 9 the error bars of the density seems quite the same

Fig. 10 error bars hard to detect -> thicker Lines

Fig. 11 images not the same size

Fig. 12 a+b values for 0 Zrb2 are half hidden -> extend to -0.5 in x axis

Author Response

Fig. 7 and 8. same description where is the difference?

The descripition was corerected.

Fig. 7 presents the morphology of phases distribution in steel-ZrB2 composites. Fig.8 presents of nickel-rich precipitates which were observed for all steel-ZrB2 composites.

ZrB2 no subscript of 2 in all figure descriptions also for Al2O3

It was corrected.

Fig. 9 the error bars of the density seems quite the same

The figure was corrected.

Fig. 10 error bars hard to detect -> thicker Lines

The figure was corrected.

Fig. 11 images not the same size

The images were corrected.

Fig. 12 a+b values for 0 Zrb2 are half hidden -> extend to -0.5 in x axis

The figures were corrected.